

# Antibacterial and anti-adherence effects of a plant extract mixture (PEM) and its individual constituent extracts (*Psidium sp.*, *Mangifera sp.*, and *Mentha sp.*) on single- and dual-species biofilms

Zaleha Shafiei[1,2], Zubaidah Haji Abdul Rahim[1], Koshy Philip[3] and Nalina Thurairajah[4]

[1] Department of Oral Biology and Biomedical Sciences, Faculty of Dentistry, University of Malaya, Kuala Lumpur, Malaysia
[2] Department of Clinical Oral Biology, Faculty of Dentistry, Universiti Kebangsaan Malaysia, Kuala Lumpur, Malaysia
[3] Institute of Biological Sciences, Faculty of Science, University of Malaya, Kuala Lumpur, Malaysia
[4] Centre for Pre-U Studies, UCSI University, Kuala Lumpur, Malaysia

Corresponding author
Koshy Philip, kphil@um.edu.my

## ABSTRACT

**Background.** Plant extracts mixture (PEM) and its individual constituent plant extracts (*Psidium* sp., *Mangifera* sp., *Mentha* sp.) are known to have an anti-adhering effect towards oral bacteria in the single-species biofilm. To date, the adhering ability of the early and late plaque colonisers (*Streptococcus sanguinis* and *Streptococcus mutans*) to PEM-treated experimental pellicle have not been investigated in dual-species biofilms.
**Methods.** Fresh leaves of these plants were used in the preparation of the respective aqueous extract decoctions. The minimum inhibitory concentration (MIC) of the extracts towards *S. sanguinis* ATCC BAA-1455 and *S. mutans* ATCC 25175 was determined using a two-fold serial microdilution method. The sum of fractional inhibitory concentration ($\Sigma$FIC) index of PEM and its constituent plant extracts was calculated using the MIC values of the plants. The minimum bactericidal concentration (MBC) of the plant extracts was also determined. The anti-adherence effect of the plant extracts (individually and mixed) was carried out by developing simulated *S. sanguinis* and *S. mutans* respectively in single- and dual-species of biofilms in the Nordini's Artificial Mouth (NAM) model system in which the experimental pellicle was pretreated with the plant extract before bacterial inoculation. The bacterial population in the respective biofilms was quantified using ten-fold serial dilutions method and expressed as colony forming unit per ml (CFU/ml). The bacterial population was also viewed using Scanning Electron Microscope (SEM). All experiments were done in triplicate.
**Results.** The PEM compared with its respective constituent plants showed the lowest MIC towards *S. sanguinis* (3.81 mg/ml) and *S. mutans* (1.91 mg/ml) and exhibited a synergistic effect. The *Psidium* sp. (15.24 mg/ml) and, PEM and *Psidium* sp. (30.48 mg/ml) showed the lowest MBC towards *S. sanguinis* and *S. mutans* respectively. The anti-adherence effect of the PEM and its respective constituent plants (except *Psidium* sp.) was different for the two bacteria in the single-species biofilm. In the dual-species biofilms, PEM demonstrated similar anti-adherence effect towards

*S. sanguinis* and *S. mutans*. The proportions of the bacterial population viewed under SEM appeared to be in agreement with the quantified population.

**Discussion**. The combination of the active constituents of the individual plant extracts in PEM may contribute to its low MIC giving rise to the synergistic effect. The different anti-adherence effect towards *S. sanguinis* and *S. mutans* in both single- and dual-species biofilms could be due to the different proportion of the active constituents of the extracts and the interaction between different bacteria. The better adhering ability of *S. sanguinis* towards the PEM-treated pellicle when present together with *S. mutans* in the dual-species biofilms may suggest the potential of PEM in controlling the balance between the early and late colonisers in biofilms.

## INTRODUCTION

Bacterial adherence has a central role in the pathogenesis of oral diseases such as dental caries and periodontal diseases, which are also known as plaque-related diseases. The formation of dental plaque or oral biofilm is as follows. After tooth brushing, the tooth surface is immediately covered with saliva forming an acellular layer known as acquired pellicle. The acquired pellicle provides receptors for the oral bacteria to adhere and colonise. The pioneer bacteria that adhere to and subsequently colonise the acquired pellicle are known as early colonisers which include *Streptococcus sanguinis* (*S. sanguinis*), *Streptococcus mitis* (*S. mitis*) and *Actinomyces* sp. (*Fathilah, Othman & Rahim, 1999*). It has been reported that *Streptococcus mutans* (*S. mutans*), the late settlers of the acquired pellicle are the causative agent of dental caries (*Bowen, 2002*). Bacteria in the oral cavity form a balanced population known as homeostasis and if imbalanced can lead to the initiation of oral diseases (*Marsh & Martin, 2009*). Inhibition of bacterial adherence to the acquired pellicle may play a role in the prevention of oral diseases.

Fluoride and chlorhexidine are among the chemicals incorporated into oral healthcare products for the prevention of dental caries. They possess bactericidal activities and demonstrate many adverse effects such as vomiting, diarrhoea, taste disturbance and teeth staining (*Chen & Wang, 2010*; *Palombo, 2011*).

Past studies have been reporting plants with a wide range of biological activities that could be useful for the development of alternative or adjunctive anti-plaque and anticaries therapies. *Psidium guajava* L. (Myrtaceae) (*P. guajava*) contains phenolic compounds that have multiple biological effects attributing to its antimicrobial and anticariogenic potentials (*Prabu, Gnanamani & Sadulla, 2006*). Its leaves decoction have long been used in folklore practices to maintain oral hygiene by exhibiting positive anti-adherence towards the adhesion of the early colonisers onto saliva-coated glass surfaces (*Fathilah & Rahim, 2003*). The extracts have the ability to alter and disturb the surface characteristics of the early plaque settlers thus reducing their adherence (*Fathilah, Othman & Rahim, 2006*).

*Mangifera indica* L. (Anacardiaceae) (mango plant) (bark, roots and leaves) have been used in traditional medicine. Its leaves extract can cause significant reduction of *Prevotella intermedia* and *Porphyromonas gingivalis* compared to toothbrushing, a home care hygiene device (*Bairy et al., 2002*). *Mentha piperita* L. (Lamiaceae) leaves extract contains tannin and flavonoids with antibacterial and antifungal activities against selected oral pathogens. Its regular intake can ward off the initial colonisation of pathogenic microbes (*Pramila et al., 2012*).

It has been shown *in vitro* that a mixture of aqueous extracts of three plant species (*Psidium* sp., *Mangifera* sp. and *Mentha* sp.) which is referred as Plant Extract Mixture (PEM) in this study has an anti-adherence effect towards early plaque colonisers (*Nordini, Fathilah & Rahim, 2013*) and towards early and late colonisers (*Rahim et al., 2014*) in single-species biofilm. To date, there is no study on the effect of PEM and its individual extracts on dual-species biofilms in a dynamic environment. Thus, this study investigated the effects of PEM and its individual constituent plant extracts towards the adherence of bacteria in single- and dual-species biofilms.

## MATERIALS AND METHODS

### Plant collection and extraction

Fresh leaves of *Psidium* sp. (voucher no. 48126) and *Mangifera* sp. (voucher no. 48124) were obtained from Balai Ungku Aziz of the University of Malaya, Kuala Lumpur and the UPM Agriculture Park, Selangor respectively. Leaves of *Mentha* sp. (voucher no. 48127) grown in Cameron Highlands, Pahang, were obtained from a local market in Selangor, Malaysia. The plants were identified at the Rimba Ilmu Herbarium (University of Malaya) and deposited under the stated voucher numbers. The fresh plant leaves were washed with tap water, followed by deionised water, dried using tissue paper, weighed and cut into small pieces before boiling based on the method described by *Nalina & Rahim (2006)*. Briefly, 1 g of fresh leaves was boiled in 1 L of deionised water for several hours until the final volume is one-third of the initial volume. The decoction was then filtered using a muslin cloth to remove any debris and the clear filtrate was further centrifuged at $1,500 \times g$, $4\,°C$ for 15 min to eliminate any sediment. The supernatants were filtered using Whatman No. 1 paper with a diameter of 150 cm and boiled again until the final volume of 100 ml. The supernatants were then frozen overnight at $-80\,°C$ followed by freeze-drying for 2 days using freeze dryer (Eyela FDV-1200, China) in a sterile environment. The sterile dried crude aqueous extracts were stored at $-20\,°C$ for further use.

### Preparation of bacterial suspension

*S. sanguinis* ATCC BAA-1455 (strain SK36) and *S. mutans* ATCC 25175 used in this study were obtained from American Type Culture Collection (ATCC, USA). Prior to the experiment, the respective 20% glycerol stocks of each bacterium at $-80\,°C$ were thawed at room temperature and 1% of the strains were transferred in 20 ml of sterile fresh Brain Heart Infusion (BHI) broth (Oxoid Ltd, Basingstoke, Hampshire, UK), incubated aerobically at $37\,°C$ with shaking at 150 rpm until mid-log phase of growth (6–8 h). The bacterial suspensions were centrifuged at $5,800 \times g$, $4\,°C$ for 10 min and the pellets were

washed three times with ice-cold sterile deionised water, suspended in fresh BHI broth and incubated at 37 °C for 15 min to reactivate their growth phase. The turbidity of each strain was standardised by adjusting the absorbance to 0.144 (equivalent to $1.00 \times 10^8$ CFU/ml and $1.53 \times 10^7$ CFU/ml for *S. sanguinis* and *S. mutans* respectively) at 550 nm using a spectrophotometer (Shimadzu UV-1700, Japan). For dual-species, the standardised suspensions of the two bacteria were mixed (1:1 vol/vol) prior to the experiment. The purity of the bacterial cultures was checked each time prior to every experiment by streaking the culture broth on BHI agar and incubated at 37 °C for 24–48 h.

## Determination of minimum inhibitory concentrations (MIC), minimum bactericidal concentrations (MBC) and the sum of the fractional inhibitory concentration (ΣFIC) index

A broth microdilution method in accordance with standard protocol CLSI (*CLSI, 2012*) and modified method of *Kuete et al. (2009)* were used to determine the MIC of PEM and its individual constituent plant extracts (*Psidium* sp., *Mangifera* sp. and *Mentha* sp.). About 1 g of sterile dried extract (*Psidium* sp.) was reconstituted in 7.81 ml sterile deionised water giving a final concentration of extract at 128 mg/ml. After sonication for 30 s and vortexing for 30 min, the reconstituted extract was centrifuged at $1,500 \times g$, 4 °C for 15 min. The supernatant of the reconstituted extract was serially diluted two-fold in BHI broth to obtain a range of concentration (0.25–64 mg/ml) in respective test tubes labelled as T1–T9. The diluted extracts were transferred into the corresponding wells of a 96-well microtitre plate (NUNC[TM] Brand products) as follows. To the first three rows of the 96-well microtitre plate labelled as A1–A9; B1–B9; C1–C9 (carried out in triplicate to ensure reproducibility), 100 μl of the respective serially diluted plant extract in BHI broth (T1–T9) was added. To wells labelled as A11, B11, C11, 100 μl of BHI broth was added to ensure bacterial growth. To all of the wells above, 5 μl of *S. sanguinis* suspension (standardised to $10^6$ CFU/ml) was added. This gave the final concentration of the extract in the respective wells (A1–A9; B1–B9; C1–C9) a range of concentration from 0.24–60.95 mg/ml. In addition to these, wells labelled as A12, B12, C12 was filled with 105 μl of BHI broth just to ensure the sterility of the broth. To another row of wells, one row apart from the others (E1–E9), 105 μl of the two-fold serially diluted plant extract only was added which served as blank control. The microtitre plate was sealed and covered with aluminium foil, incubated aerobically at 37 °C with shaking at 150 rpm for 24 h. Following incubation, the MIC was determined at 550 nm using ELISA reader based on turbidity determined by comparing the absorbance of the suspension in wells of test extract (A1–A9; B1–B9; C1–C9) with that of the corresponding blank control. The MIC of each compound was defined as the lowest concentration that inhibited the bacterial growth. Similar procedure was repeated with *S. mutans*. The similar experiment was repeated for the other plant extracts (*Mangifera* sp., *Mentha* sp. and PEM). For PEM, an equal volume of the stock solution of extracts (*Psidium* sp., *Mangifera* sp. and *Mentha* sp.) each with a concentration of 128 mg/ml was mixed followed by two-fold serial dilution.

For the MBC determination, 20 μl of the bacterial suspension from selected wells (A1–A9; B1–B9; C1–C9) which showed almost no turbidity was inoculated on BHI agar

(Oxoid Ltd, Basingstoke, Hampshire, UK) and incubated for 48 h at 37 °C. The lowest concentration of the extracts that showed no growth corresponded to the MBC value.

The sum of the fractional inhibitory concentration ($\Sigma$FIC) determines the synergy of the plant extracts, calculated using the following equation:

$$\Sigma FIC = 1/3(FIC_I + FIC_{II} + FIC_{III})$$

and the FICs were calculated as follows:

$$FIC_I \text{ (FIC of \textit{Psidium} sp.)} = \frac{MIC \text{ (PEM)}}{MIC \text{ (\textit{Psidium} sp.)}}$$

$$FIC_{II} \text{ (FIC of \textit{Mangifera} sp.)} = \frac{MIC \text{ (PEM)}}{MIC \text{ (\textit{Mangifera} sp.)}}$$

$$FIC_{III} \text{ (FIC of \textit{Mentha} sp.)} = \frac{MIC \text{ (PEM)}}{MIC \text{ (\textit{Mentha} sp.)}}.$$

The $\Sigma$FIC index determines the interaction between the different plant extracts in the mixture where the interaction is interpreted according to a range of values; value less than 0.5 as synergistic, value greater than 0.5–1 as additive, value greater than 1–4 as indifferent and value greater than 4 as antagonistic (*Van Vuuren & Viljoen, 2011*).

## Preparation of sterilised saliva

Undiluted sterile saliva was prepared according to the method described by *De Jong & Van der Hoeven (1987)* and *Nordini, Fathilah & Rahim (2013)*. To minimize variation, a single volunteer was used to collect 100 ml of stimulated whole saliva (SWS). The SWS collection was done by expectoration after chewing sugar-free gum (before eating or at least 2 h after eating) in ice-chilled test tubes. The aggregation of the protein in the SWS was minimised by adding 1,4-Dithio-DL-threitol (DTT) (GE Healthcare, Danderyd, Sweden) to a final concentration of 2.5 mM. The mixture was stirred slowly for 10 min followed by centrifugation at 800 × g, 4 °C for 30 min. The supernatant was then filter sterilised through a disposable 0.22 μm low-protein-binding filter (Cellulose acetate syringe filters Sartorius, USA) into sterile centrifuge tubes and stored at −20 °C until further use. Prior to the experiment, the sterile SWS was thawed and centrifuged to remove any precipitate It was later used to coat glass beads (3 mm in diameter) (Merck, Damstadt, Germany) in an artificial mouth (NAM) model forming a layer that mimics the acquired pellicle on the tooth surface and thus referred as the Experimental Pellicle.

## Ethical approval

The study protocol was reviewed and approved by the Ethics committee of Faculty of Dentistry University of Malaya, Kuala Lumpur, Malaysia (The Ethic committee/IRB reference number: DF OB1506/0070(P). Written informed consent was obtained from one donor for saliva collection.

## Anti-adherence effect of plant extracts

Prior to use, the dried aqueous extracts were reconstituted in sterile deionised water to a final concentration of 0.5 mg/ml. The reconstituted extracts were further sterilised by

filtration using 0.22 μm nylon syringe filters (Millipore, Billerica, MA, USA). For PEM, the same concentrations of the sterile reconstituted individual extracts were mixed (1:1:1 vol/vol/vol) to give a mixture of extracts of the same concentration (0.5 mg/ml).

Nordini's Artificial Mouth (NAM) model developed by *Rahim et al. (2008)* was used in the *in vitro* study of the anti-adherence effect of PEM and its individual constituent plant extracts towards the early and late plaque colonisers in single- and dual-species biofilms. The NAM model was used in the study to represent oral cavity. The model consists of a glass chamber (1 × 6 cm length), glass beads (3 mm diameter), water bath system, saliva reservoir, bacterial reservoir and a peristaltic pump. In this study, nine glass beads were placed in the glass chamber of the NAM model. The glass beads represent the tooth surface. The glass chamber which was immersed in the water bath system, set at 37 °C represents the oral cavity. The peristaltic pump (Masterflex L/S, USA) of the NAM model was used to flow the saliva/test extract/bacteria from the reservoir into the system via rubber tubing at a constant flow rate (0.3 ml/min for resting saliva). The experiment was carried out as follows. Saliva was first flowed into the NAM model for 2 min at a constant rate to form an experimental pellicle on the glass beads. Excess saliva was rinsed off by allowing sterile deionised water to flow in for 2 min. This allowed the experimental pellicle to be pretreated with sterile deionised water. Following this, *S. sanguinis* which was standardised to 0.144 absorbance at 550 nm wavelength was allowed to flow into the NAM model for 24 h to form a 24-h biofilm on the experimental pellicle-coated glass beads.

Of the nine glass beads, three were taken out randomly whereas the remaining six glass beads were kept for SEM viewing and reserves. The three glass beads were then individually placed in a sterile microcentrifuge tube containing 1 ml Phosphate Buffered Saline (PBS) (pH 7.4), sonicated for 10 s and vortexed for one minute to dislodge the attached bacteria. The suspension was serially diluted ten-fold ($10^{-1}$–$10^{-5}$ dilutions) with PBS. A 100 μl of the homogeneous bacterial suspension from each tube was pipetted out and plated on three separate BHI agar plates using the Lawn method. The plates were incubated for 24 h at 37 °C. The counting of all viable microorganisms formed on the plate done using colony counter. Plates from the dilution which gave a CFU number of between 30–300 colonies were selected and used in the calculation of the growth population. The bacterial suspension used in the plating was 100 μl and hence it was expressed as CFU/ml based on the following formula:

$$\text{Total CFU/ml} = \frac{\text{Number of colonies}}{\text{Volume plated (ml)} \times \text{Dilution factor}}.$$

The deionized water treated experimental pellicle served as negative control for *S. sanguinis* in single-species biofilm and the bacterial population (total CFU/ml) assumed as 100% bacterial adherence. Similar procedure was repeated with *S. mutans* (standardised to 0.144 absorbance at 550 nm) for single-species biofilm. For dual-species biofilms, the individual strain was mixed in equal amount (1:1 vol/vol) and a similar procedure was repeated. In the dual-species biofilms, the two bacteria were differentiated according to their colony sizes. *S. mutans* colony is larger than *S. sanguinis* allowing the colonies to be counted separately.

The experiment was subsequently repeated where the experimental pellicle was pretreated with respective plant extract (*Psidium* sp./ *Mangifera* sp./ *Mentha* sp./PEM) at sub-MIC (0.5 mg/ml) before inoculation with bacterial suspension used for the development of single- and dual-species biofilms. For positive control, the experimental pellicle was pretreated with 0.12% chlorhexidine gluconate (CHX). The individual experiment was carried out in triplicate.

Percentage of adherence (X) was calculated using the equation below:

$$X = \frac{\text{adhered cells (test or positive control)} \times 100}{\text{adhered cells (negative control)}}.$$

Assuming the bacterial adherence to negative control was 100%, the percentage of anti-adherence was calculated as:

$$\%\text{Anti-adherence} = 100 - X.$$

## Scanning Electron Microscopy (SEM) analysis of cell population on glass beads

The remaining beads from the experiments were processed for SEM viewing according to the method described by *Rahim & Thurairajah (2011)* with slight modification. Briefly, three glass beads for each of experiment were fixed with 1 ml of 4% glutaraldehyde solution (prepared in sodium cacodylate buffer, pH 7.4) and kept at 4 °C until the subsequent analysis. Prior the analysis, 4% glutaraldehyde solution was pipetted out and washed twice with sodium cacodylate buffer for 15 min, followed by postfixing with 2% (vol/vol) osmium tetroxide in 1% (wt/vol) sodium cacodylate buffer solution for 1 hour and kept at 4 °C. Then, the glass beads were rinsed with deionised water twice for 15 min before dehydration process. A series of 30, 50, 70, 80, and 95% ethanol at an interval of 15 min was used for dehydration process. This was followed by dehydration carried out twice for 15 min each using 100% ethanol. Ethanol was gradually displaced with acetone in the following ratios (vol/vol); Ethanol:Acetone 3:1, 1:1, 1:3 each for 20 min and finally followed by 100% acetone three times for 20 min. The dehydrated samples were then subjected to Critical Point Drying (CPD) (Balzers CPD 030; Balzers Union, Balzers, Liechtenstein) for 1 h 40 min in liquid $CO_2$ under 95 bar pressure. The glass beads were then kept in a tight container in a desiccator. Prior to SEM viewing, the beads were gold-coated under low pressure with ion sputter coater (Joel JFC1100, Japan). The beads were viewed for cell population at 10,000X magnification using Scanning Electron Microscope (Quanta FEG 250; FEI, Eindhoven, The Netherlands).

## Statistical analysis

The experiments were repeated and analysed using three triplicates and expressed as a mean ± standard deviation, where the number of determinations, $n = 27$. Data were analysed using IBM SPSS statistical software, version 23. The Shapiro–Wilk test was used to test assumptions of normality. The results were analysed using Kruskal–Wallis and Mann–Whitney tests for non-parametric statistical analysis to compare the difference between the two independent groups. Results were considered significant at $p < 0.05$.

**Table 1** **The MIC, MBC and FIC value of PEM and its individual constituent plant extracts against tested oral bacteria.**

| | [a]MIC(MBC) (mg/ml) | | | | Fractional inhibitory concentration index | | | | |
| | *Psidium* sp. | *Mangifera* sp. | *Mentha* sp. | [b]PEM | $FIC_I$ | $FIC_{II}$ | $FIC_{III}$ | [c]$\Sigma FIC = 1/3$ ($FIC_I$ $+ FIC_{II} + FIC_{III}$) | Interaction |
|---|---|---|---|---|---|---|---|---|---|
| *S. sanguinis* ATCC BAA-1455 | 7.62 (15.24) | 15.24 (30.48) | 7.62 (30.48) | 3.81 (30.48) | 0.50 | 0.25 | 0.50 | 1/3(0.50 + 0.25 + 0.50) = 0.42 | Synergistic |
| *S. mutans* ATCC 25175 | 3.81 (30.48) | 15.24 (60.95) | 15.24 (60.95) | 1.91 (30.48) | 0.50 | 0.13 | 0.13 | 1/3(0.50 + 0.13 + 0.13) = 0.25 | Synergistic |

**Notes.**

The results are shown as average values of triplicate.

[a]MIC(MBC): minimum inhibitory concentration (minimum bactericidal concentration).

[b]PEM: Plant extract mixture of *Psidium* sp., *Mangifera* sp. and *Mentha* sp. in ratio of 1:1:1 ($w/v$).

[c]$\Sigma FIC$: the sum of fractional inhibitory concentration and was calculated according to the formula, $\Sigma FIC = 1/3(FIC_I + FIC_{II} + FIC_{III})$.

The $\Sigma FIC$ index was interpreted as interactions which are synergistic (if < 0.5), additive (if in the range of > 0.5–1) and indifferent (if in the range > 1–4), or as antagonist (if > 4).

## RESULTS

### Minimum inhibitory concentrations (MIC), minimum bactericidal concentrations (MBC) and the sum of the fractional inhibitory concentration ($\Sigma FIC$) index of plant extracts

Table 1 shows the MIC, MBC and $\Sigma FIC$ index of the PEM and its individual constituent plant extracts against the early (*S. sanguinis*) and late (*S. mutans*) plaque colonisers. The MIC values of PEM (*S. sanguinis* 3.81 mg/ml; *S. mutans* 1.91 mg/ml) were lower than those of the individual extracts for the two bacteria. Among its individual constituent plant extracts, the *Psidium* sp. showed the lowest MIC against *S. mutans* (3.81 mg/ml) whereas *Mentha* sp. and *Psidium* sp. have almost similar MIC against *S. sanguinis* (7.62 mg/ml). It is shown that the MBC were higher than the MIC for all the extracts. *Psidium* sp. (MBC value of 15.24 mg/ml) showed the higher bactericidal effect compared with *Mangifera* sp., *Mentha* sp. and PEM (MBC value of 30.48 mg/ml) against *S. sanguinis*. The MBC value exhibited by PEM and *Psidium* sp. against *S. mutans* (30.48 mg/ml) was lower compared with *Mangifera* sp. and *Mentha* sp. (60.95 mg/ml). It was found from the calculation of the $\Sigma FIC$ index, the interactions of the plant extracts are synergistic for the two bacteria where the index is lower for *S. mutans* ($\Sigma FIC = 0.25$) compared with *S. sanguinis* ($\Sigma FIC = 0.42$).

### Bacterial population adhering to the experimental pellicle

The bacterial population for *S. sanguinis* and *S. mutans* adhering to experimental pellicle without treatment (negative control) either individually (single-species) or together (dual-species) is shown in Table 2. In a single-species biofilm, the population of *S. mutans* (($2.48 \pm 0.16$) $\times 10^7$ CFU/ml) adhering to the experimental pellicle was ten times higher compared with *S. sanguinis* (($2.30 \pm 0.37$) $\times 10^6$ CFU/ml) and it is statistically significant ($p < 0.05$). However, in dual-species biofilms, the *S. mutans* population (($1.54 \pm 0.35$) $\times 10^7$ CFU/ml) adhered in similar proportions with *S. sanguinis* (($1.49 \pm 0.36$) $\times 10^7$ CFU/ml).

**Table 2  Bacterial population in CFU/ml for the negative control for *S. sanguinis* and *S. mutans* singly and in the mixture.**

| Bacteria | Bacteria population adhered to the experimental pellicle (negative control)[a] (mean ± SD) (CFU/ml) |
|---|---|
| *S. sanguinis* ATCC BAA-1455 | $(2.30 \pm 0.37) \times 10^{6*}$ |
| *S. mutans* ATCC 25175 | $(2.48 \pm 0.16) \times 10^{7*}$ |
| *S. sanguinis* ATCC BAA-1455 + *S. mutans* ATCC 25175 | $(1.49 \pm 0.36) \times 10^{7} + (1.54 \pm 0.35) \times 10^{7}$ |

**Notes.**
The results are shown as average values of triplicate ($n = 27$) and are expressed in a mean CFU/ml ± standard deviation (SD).
[a]Negative control represents bacteria population adhered to the experimental pellicle-coated glass beads and was assumed as 100%.
*$p < 0.05$ considered as significant.

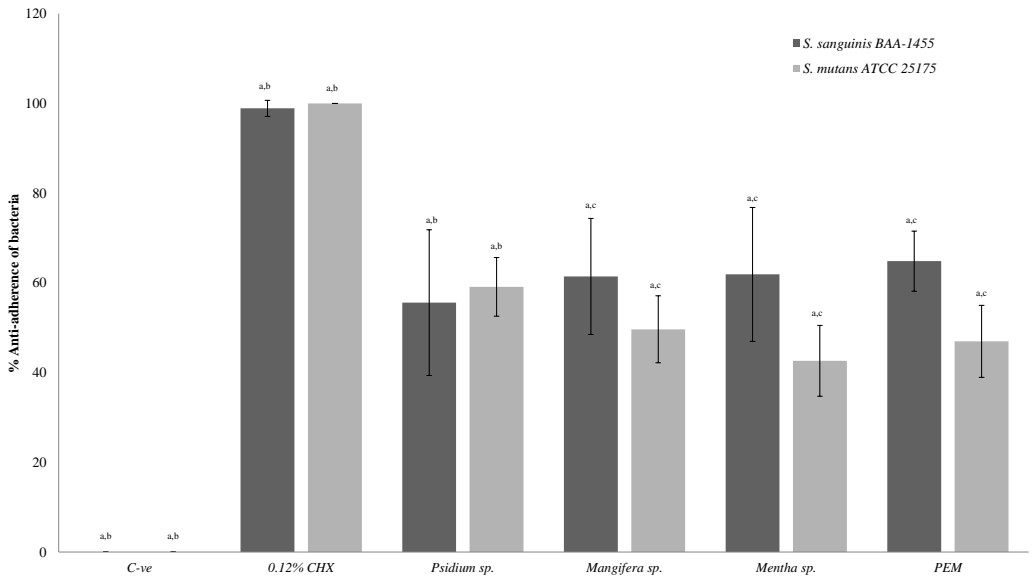

**Figure 1  The anti-adherence effect of PEM and its respective plant extracts in single-species biofilm (*S. sanguinis* and *S. mutans*).** The percentage of anti-adhered bacterial cells present in the biofilm was expressed as mean ± standard deviation (SD) where a number of determinations ($n$) = 27. a-$p < 0.05$ comparing between PEM, *Psidium* sp., *Mangifera* sp., *Mentha* sp., 0.12% CHX (positive control) and negative control (Kruskal–Wallis test). b-$p < 0.05$ comparing between PEM and *Psidium* sp., PEM and positive control and PEM and negative control (Mann–Whitney test). c-$p < 0.05$ comparing between *S. sanguinis* and *S. mutans* (treated with *Mangifera* sp.), *S. sanguinis* and *S. mutans* (treated with *Mentha* sp.) and *S. sanguinis* and *S. mutans* (treated with PEM) (Mann–Whitney test).

## Anti-adherence effect of plant extracts in single- and dual-species biofilms

Figure 1 shows the anti-adherence effects of PEM and its respective constituent plant extracts in single-species biofilm (*S. sanguinis* and *S. mutans*). The anti-adhering effect of all the extracts (except *Psidium* sp.) towards *S. sanguinis* was significantly higher when compared with *S. mutans*.

Figure 2 shows the anti-adherence activities of the PEM and its individual constituent plant extracts in the dual-species biofilms. It was observed that *Psidium* sp. also

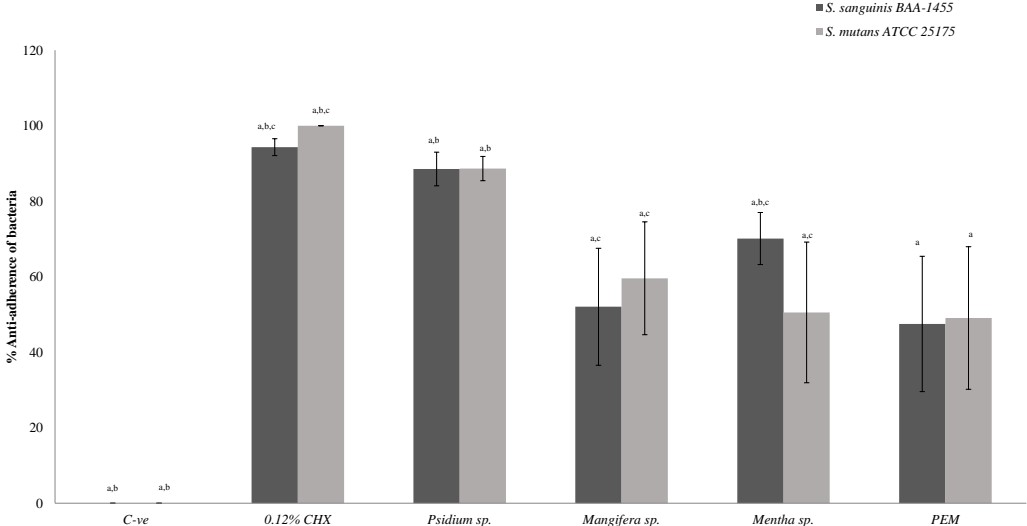

**Figure 2** **The anti-adherence effect of PEM and its respective plant extract on dual-species (*S. sanguinis* + *S. mutans*) biofilms.** The percentage of anti-adhered bacterial cells present in the biofilm was expressed as mean ± standard deviation (SD) where a number of determinations (*n*) = 27. a-*p* < 0.05 comparing between the respective PEM, *Psidium* sp., *Mangifera* sp., *Mentha* sp., 0.12% CHX (positive control) and the negative control (Kruskal–Wallis test). b-*p* < 0.05 comparing between PEM and *Psidium* sp., PEM and *Mentha* sp. (for *S. sanguinis*), PEM and positive control and, PEM and negative control (Mann–Whitney test). c-*p* < 0.05 comparing between *S. mutans* and *S. sanguinis* (treated with *Mangifera* sp.), *S. mutans* and *S. sanguinis* (treated with *Mentha* sp.), *S. mutans* and *S. sanguinis* (treated with positive control) and, *S. mutans* and *S. sanguinis* (treated with negative control) (Mann–Whitney test).

demonstrated similar and higher anti-adherence activities towards *S. sanguinis* (88.54 ± 4.47%) and *S. mutans* (88.65 ± 3.22%) compared with PEM, *Mangifera* sp. and *Mentha* sp. The *Mentha* sp. exhibited significantly higher anti-adherence activity towards *S. sanguinis* (70.12 ± 6.90%) compared with *S. mutans* (50.54 ± 18.64%). PEM demonstrated comparable anti-adherence effect towards *S. sanguinis* and *S. mutans* (47.50 ± 17.93% and 49.08 ± 18.90% respectively).

## Bacterial adherence to plant extracts treated-experimental pellicle viewed under SEM

Figures 3 and 4 show the SEM photographs of single-species biofilm viewed at 10,000X magnification displaying the cell population adhering to the experimental pellicle (saliva-coated glass beads) treated with plant extracts as well those of the negative and positive controls. The population of *S. mutans* and *S. sanguinis* were almost zero in single-species biofilm adhering to 0.12% CHX-treated experimental pellicle (positive control) and very much reduced with plant extract-treated when compared with the negative control.

Figure 5 shows the SEM photographs of the dual-species biofilms. It was observed that the population of the two bacteria adhering to *Psidium* sp.-treated experimental pellicle was highly reduced (Fig. 5D). It was different for *Mentha* sp-treated experimental pellicle, the adhered population of *S. sanguinis* was less compared with *S. mutans* (Fig. 5F). The adherence of the two bacteria to *Mangifera* sp.- (Fig. 5E) and PEM-treated experimental pellicle (Fig. 5C) respectively were almost comparable.

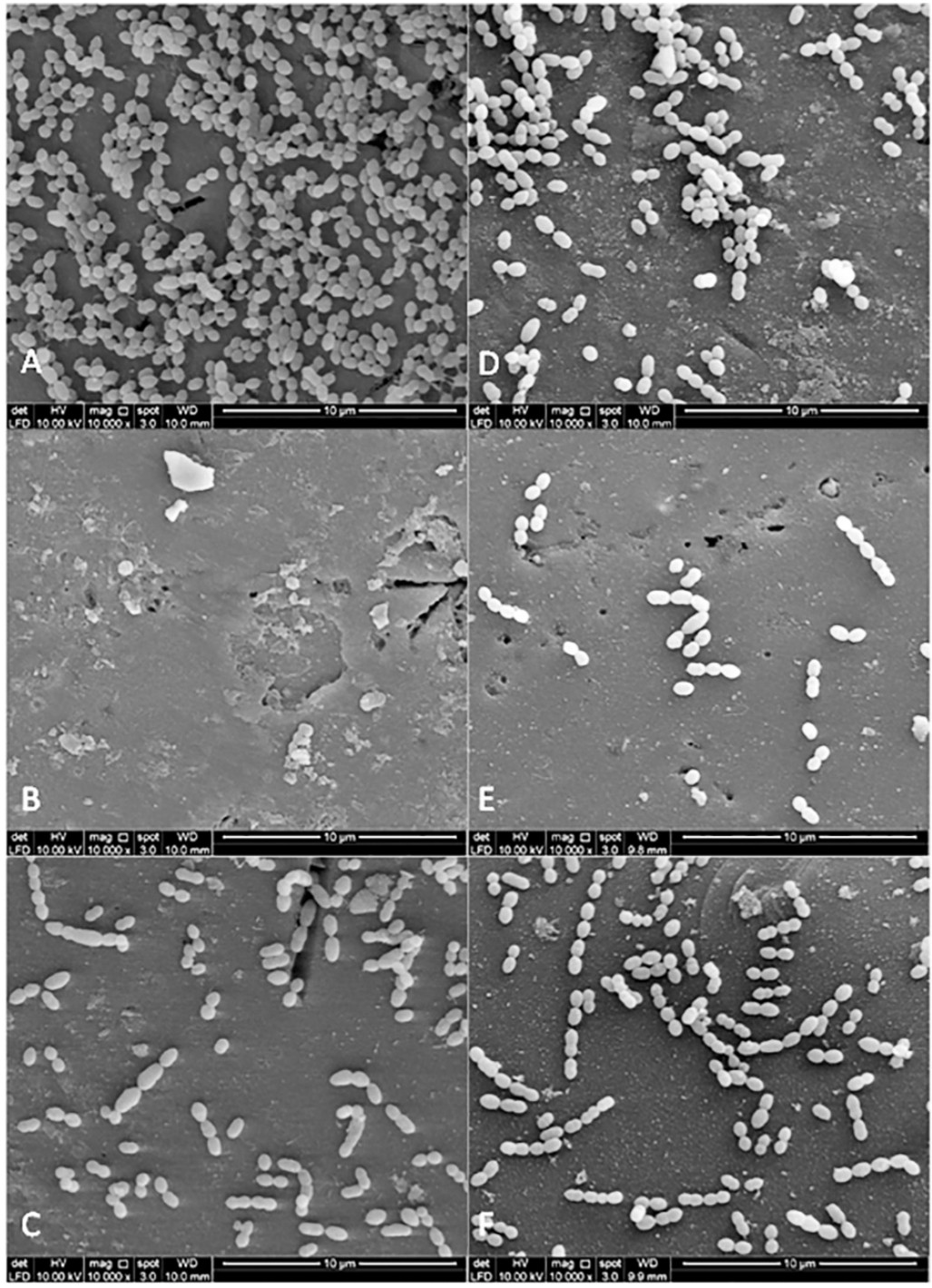

**Figure 3** **The cell population of *S. sanguinis* viewed by Scanning Electron Microscopy (SEM) where the pellicle was treated with PEM (C) and its respective plant extracts (*Psidium* sp (D); *Mangifera* sp. (E) and *Mentha* sp. (F)).** The 0.12% chlorhexidine gluconate (B) and deionised water (A) served as positive and negative controls respectively. Magnification: 10,000X.

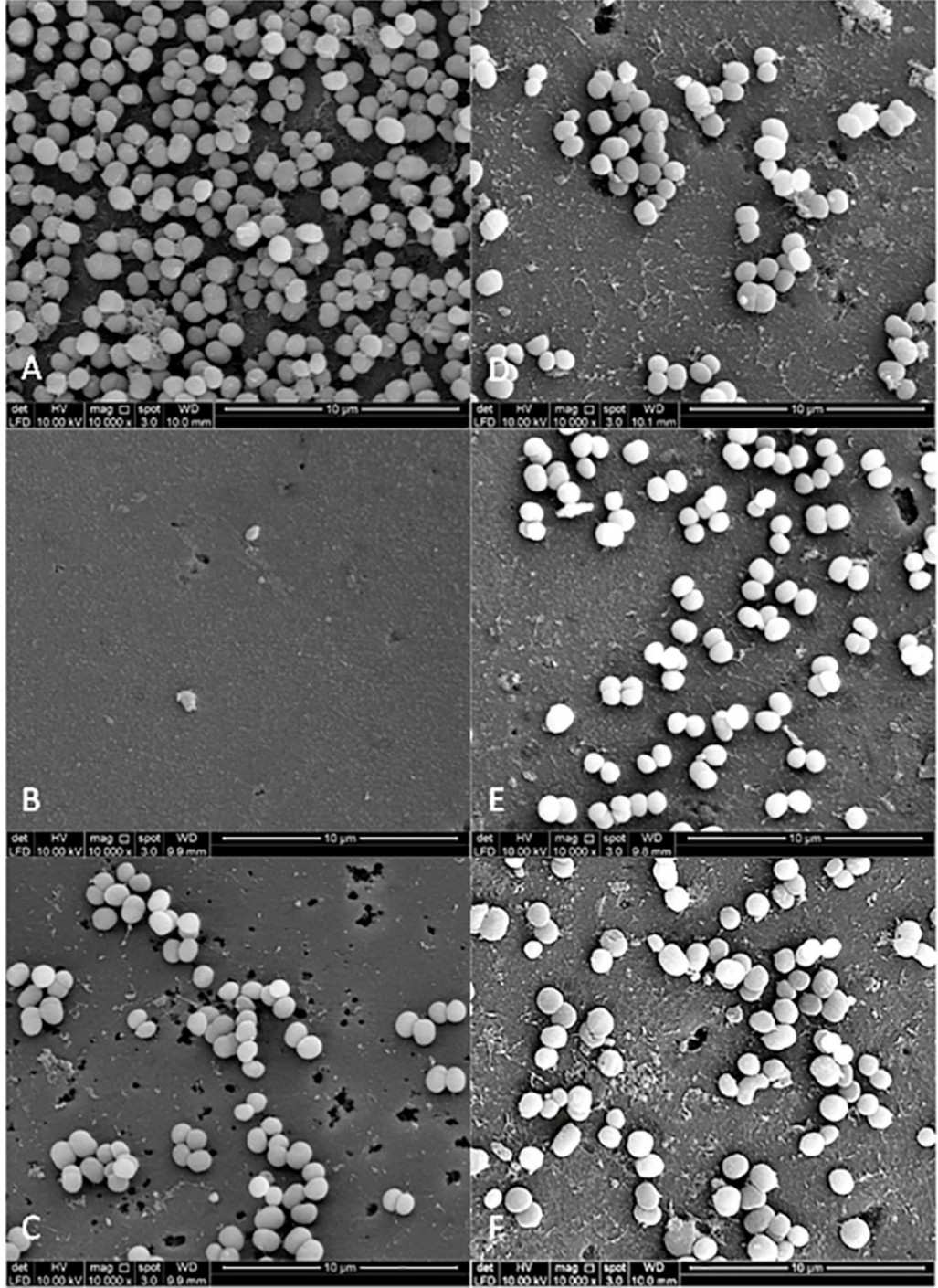

**Figure 4** **The cell population of *S. mutans* viewed by Scanning Electron Microscopy (SEM) where the pellicle was treated with PEM (C) and its respective plant extracts (*Psidium* sp (D); *Mangifera* sp. (E) and *Mentha* sp. (F)).** The 0.12% chlorhexidine gluconate (B) and deionised water (A) were served as positive and negative controls respectively. Magnification: 10,000X.

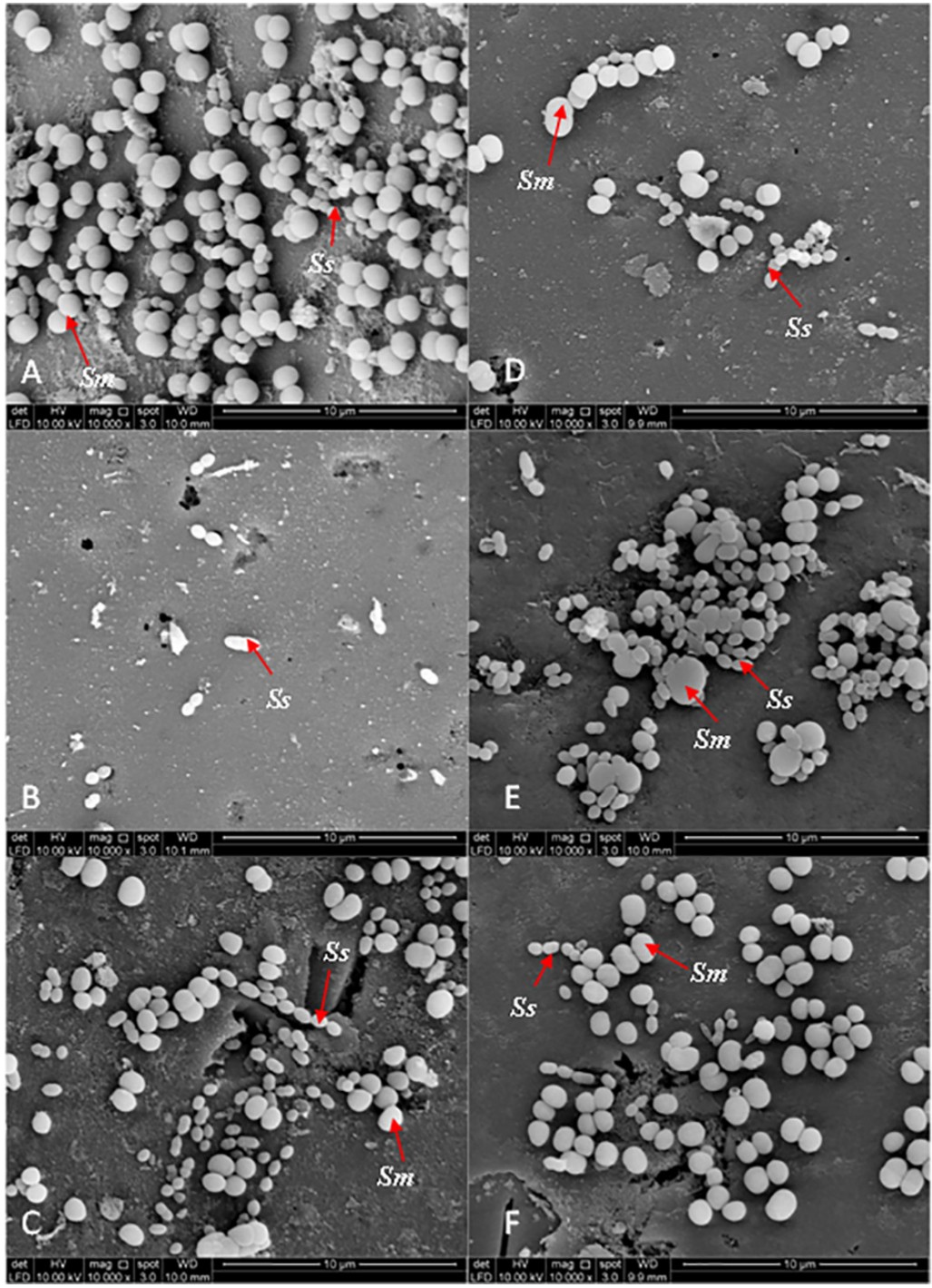

**Figure 5** **The cell population of dual-species biofilms (*S. sanguinis* (*Ss*) + *S. mutans* (*Sm*)) viewed by Scanning Electron Microscopy (SEM) where the pellicle was treated with PEM (C) and its respective plant extracts (*Psidium* sp. (D); *Mangifera* sp. (E) and *Mentha* sp. (F)) on cell adherence.** The 0.12% chlorhexidine gluconate (B) and deionised water (A) were served as positive and negative controls respectively. Magnification: 10,000X.

## DISCUSSION

In this study, the MIC value of *Psidium* sp. against *S. sanguinis* was higher (7.62 mg/ml) compared with what has been reported by *Fathilah (2011)* which is 4.69 mg/ml. The difference could be due to the bacterial strain used in which used clinical isolate whereas in this study the bacterial strain was purchased from ATCC. *Prabu, Gnanamani & Sadulla (2006)* have reported the MIC value of guaijaverin (4 mg/ml) towards *S. mutans* MTCC 1943. Guaijaverin is a biologically active flavonoid of *P. guajava* (*Prabu, Gnanamani & Sadulla, 2006*). The value obtained from our study can be considered within range as we were using the crude aqueous extract.

PEM which is a mixture of equal amount of *Psidium* sp., *Mangifera* sp. and *Mentha* sp. has low MIC compared with its individual constituent plant extracts. This suggests that PEM is a better growth inhibitor which may be attributed to the combination of the active constituents of the individual plant extracts, a synergistic interaction as indicated by the ΣFIC index.

PEM has the same MBC value towards *S. sanguinis* and *S. mutans* indicating that it exerts similar bactericidal effect against the two bacteria. However, the *Psidium* sp. has lower MBC value compared with PEM towards *S. sanguinis*. The lower MBC value might be due to *S. sanguinis* aggregating in the presence of *Psidium* sp. (*Fathilah, 2011*) as the aggregated *S. sanguinis* may not be reflected during plating. The similar MBC value of PEM towards the two bacteria indicate that PEM has component(s) that may be bactericidal to both. The synergistic effects of the individual constituent plant extracts in PEM contribute to its bactericidal and growth inhibitory effects. It has been reported that there are synergistic interactions for constituents within a total extract of a single herb, as well as between different herbs in a formulation (*Williamson, 2001*; *Nahrstedt & Butterweck, 2010*).

Bacterial adherence has a central role in the formation of dental plaque (*Jakubovics & Kolenbrander, 2010*). It is initiated by the formation of acquired pellicle on the tooth surface which then become the substratum for bacteria colonization (*Jakubovics & Kolenbrander, 2010*). Dental plaque, if not controlled, will subsequently lead to dental caries and periodontitis. It has been a normal practice to brush teeth after meals. The scenario may be different if oral healthcare products are to be used before meals. In this study, we simulate the use of healthcare product before meals by treating the experimental pellicle with plant extracts before inoculating it with the bacteria. This determines the adhering ability of bacteria towards the treated experimental pellicle. Using the plant extract at sub-MIC range (*Hasan, Danishuddin & Khan, 2015*) will allow adherence by viable cells only. The final concentration of PEM and its individual constituent plant extracts (0.5 mg/ml) that were used in this study was well below the MIC which has also been used for PEM in previous studies in single-species biofilm (*Rahim et al., 2014*).

In this study, the trend of adhered bacterial population to the experimental pellicle is different in single-species compared with dual-species biofilms. In single-species, the adhered *S. sanguinis* population was 10-fold lower compared with that of *S. mutans* This is agreement with what has been reported by *Rahim et al. (2014)*. This may suggest that
in single-species biofilm, the bacterium (*S. sanguinis* and *S. mutans* respectively) interacts independently with the components of the experimental pellicle. The behaviour of the bacterial population was found to be different when they exist together in dual-species biofilms in which the two bacteria were inoculated at the same time. The adhered bacterial population of the two bacteria was almost in equal proportion. This can be attributed to the reduced competition between the bacteria when they are inoculated at the same time as has been reported by *Kreth et al. (2005)*.

The normal adhering process of the early and late colonisers (*S. sanguinis* and *S. mutans* respectively) is disturbed when the experimental pellicle is pretreated with plant extracts before inoculation (*Rahim et al., 2014*). Their study involved single-species biofilm while our study included single- and dual-species biofilms in a dynamic environment. In a single-species biofilm, the PEM showed a higher anti-adhering activity towards *S. sanguinis* compared with *S. mutans.* The individual constituent plant extracts except *Psidium* sp. also showed the same profile of anti-adherence. The latter is not in agreement with what has been reported by *Rahim et al. (2014)* which was carried out in the same dynamic environment. The discrepancy could be due to the different strain of *S. sanguinis* used in our study. In the dual-species biofilms, the pretreatment of experimental pellicle with the plant extracts demonstrated a different profile of anti-adhering activity compared to single-species. The anti-adhering activity exhibited by the *Psidium* sp. towards the two bacteria is highest compared with the other extracts. PEM showed almost similar or about 50% of the anti-adhering activity compared with the negative controls towards the two bacteria. This suggests that the adherence of *S. mutans* to PEM-treated experimental pellicle enhanced the adherence of *S. sanguinis* when they are together. The balanced bacterial population is important to maintain oral health against the emergence of pathogenic bacteria following an imbalance in the oral resident microbiota (*Marsh & Martin, 2009*). For example, the excess population of *S. mutans* can lead to the initiation of dental caries (*Bowen, 2002*). The increase in the adherence of *S. sanguinis* in the presence of *S. mutans* as in the dual-species may be attributed to modification of the complementary binding sites between the adhesins on the bacterial surface and receptors on the treated pellicle (*Rahim et al., 2014*). The population of *S. sanguinis*, a normal oral microflora, should be maintained as it has the ability to generate antibacterial substance towards putative periodontal pathogens (*Ma et al., 2014*) where the loss of its colonization appears to be associated with aggressive periodontitis (*Stingu et al., 2008*). The colonization of *S. sanguinis* provides suitable substratum for colonization of *S. mutans* following which the population of *S. sanguinis* is reduced (*Caufield et al., 2000*; *Kreth et al., 2005*). The positive control,0.12% CHX appeared to have no adhering activity as the concentration used is more to the bactericidal (*Rahim & Thurairajah, 2011*).

The anti-adhering activity observed in plant extracts treated-experimental pellicle is a reflection of viable cell adherence since the concentration used in the study was in the sub-MIC range. The phenolic content of the plant extracts (*Thimothe et al., 2007*) may interfere the adherence of the bacterial cells to the experimental pellicle. The reduced adherence may subsequently affect the biofilm formation (*Rahim et al., 2014*; *Barnabé et al., 2014*).

The bacterial colonies adhering to the experimental pellicle (or plant extract-treated and untreated) corresponded with the bacterial population viewed under SEM confirming the effect of plant extracts. The *Psidium* sp. appears to aggregate *S. sanguinis* (Fig. 3D) and this is in agreement with what had been reported by *Fathilah (2011)*. Future studies may include the inoculation of early coloniser prior to the late coloniser for dual-species biofilms. This is to study whether the early coloniser to the plant extracts treated experimental pellicle has any influence on the adherence of the later coloniser.

## CONCLUSION

PEM appears to demonstrate a balanced population of the early-(*S. sanguinis*) and late-(*S. mutans*) plaque colonisers. This may suggest its potential use as an oral healthcare product in controlling the development of periodontitis and caries.

## ACKNOWLEDGEMENTS

I would like to acknowledge all the lab staff at Balai Ungku Aziz Research Lab for providing the place and equipment to do the study. SEM was performed at Electron Microscopy Lab, Faculty of Medicine, UM and Faculty of Dentistry, UM for processing and viewing SEM samples respectively.

### Funding

This research was supported by the Postgraduate Research Grant (PG074-2013A), Faculty of Dentistry, University of Malaya under Zubaidah Haji Abdul Rahim and University of Malaya and the High Impact Research–Malaysian Ministry of Higher Education grant designated as UM.C/625/1/HIR/MOHE/SC/08 with account number F000008-21001 under the Principal Investigator Koshy Philip. The funders had no role in study design, data collection and analysis, decision to publish, or preparation of the manuscript.

### Grant Disclosures

The following grant information was disclosed by the authors:
Postgraduate Research: PG074-2013A.
Malaysian Ministry of Higher Education: UM.C/625/1/HIR/MOHE/SC/08.

### Competing Interests

The authors declare there are no competing interests.

### Author Contributions

- Zaleha Shafiei conceived and designed the experiments, performed the experiments, analyzed the data, contributed reagents/materials/analysis tools, wrote the paper, prepared figures and/or tables, provided references.
- Zubaidah Haji Abdul Rahim conceived and designed the experiments, analyzed the data, contributed reagents/materials/analysis tools, wrote the paper, prepared figures and/or tables, reviewed drafts of the paper, editing and final.

- Koshy Philip conceived and designed the experiments, wrote the paper, reviewed drafts of the paper, proof reading and submission.
- Nalina Thurairajah analyzed the data, wrote the paper, prepared figures and/or tables, reviewed drafts of the paper, proof reading.

## Data Availability

The raw data has been supplied as Supplementary File.

## Supplemental Information

Supplemental information for this article can be found online at http://dx.doi.org/10.7717/peerj.2519#supplemental-information.

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
