# Peer review of "Antibacterial and anti-adherence effects of a plant extract mixture (PEM) and its individual constituent extracts (Psidium sp., Mangifera sp., and Mentha sp.) on single- and dual-species biofilms"

_PeerJ, doi:10.7717/peerj.2519_

## Round 0.1 · original submission · Major Revisions

· Academic Editor

Major Revisions

Dear Authors,

Expert reviewers in the field have now reviewed your paper.

Although it is of potential interest in the scientific field the paper needs more focusing. I suggest a through re-casting of the entire manuscript and to answer to all the criticisms raised.

Best regards,
Maria Rosaria

Reviewer 1 ·

Basic reporting

The structure of the article is not conform to Peer J template.

Experimental design

Methods are not described with sufficient information to be reproducible.

Validity of the findings

Data are not robust, statistically sound and controlled.

Additional comments

In this study, the authors aimed to investigate the adherence of bacteria to the experimental pellicle treated with a plant extract mixture and its constituents in the development of single- and dual-species oral biofilms.
Although the paper appears to be of potential interest in the scientific field, in my opinion, it needs more focusing and it is not apt for publication in its present form. I suggest a through re-casting of the entire manuscript.
The organization of the article is absolutely not satisfactory; this journal recommends sections in this order, Abstract, Background, Methods. Results, Discussion. The authors wrongly put the methods section at the end of the manuscript.
Lines 67-69: Rewrite this sentence, as written it makes no sense.
The results are not clearly presented, discussed and interpreted, thus the manuscript needs a careful editing so that the results of the study become clear to the reader.
Table 1 and Table 2 should be combined into one unique table.
In table 1 and 2, standard deviations are missing.
Data reported in Table 3 should be presented as logarithms, because as shown they are not clear.
Why is bacterial population expressed in CFU/ml and not in CFU/cm2? How were the two strains counted?
The method section is really poor written.
Lines 239-240: the experimental procedure should be described in detail because the method description should help the reader into understanding better the results.
Lines 223, 235, and so on: Delete commas before [..].
Figure 1 and Figure 2 should be discussed more clearly.
The reference section should be checked.

Reviewer 2 ·

Basic reporting

English language should be checked professionally, there are some mistakes.
Abstract has to be consisted from 4 parts as in the instructions for authors, so Methods should be added and there is space to do such, because Abstract can contain 500 words. The whole manuscript has to be organized in its parts in order given in templates.
In whole text the names of species, when additionally mentioned, have to be properly abbreviated, example: change from Strep. mutans to S. mutans, etc.
References in the text and in the list have to be changed according to the instructions for authors.
Correct from 11 to 12 Times New Roman where mistaken. The % symbol has to be written without space between it and any number.

Experimental design

Research is original and continues previous knowledge.

Validity of the findings

The findings are acceptable.

Additional comments

Structure the article according to the PeerJ instructions and do all the changes given as red text in the reviewed article.

Annotated reviews are not available for download in order to protect the identity of reviewers who chose to remain anonymous.

Reviewer 3 ·

Basic reporting

Paper:
Antibacterial and anti-adherence effects of a plant extract mixture (PEM) and its individual constituent extracts (Psidium sp., Mangifera sp., and Mentha sp.) on single- and dual species biofilms

by Shafiei et al., PeerJ

General comments:
the present work dealt with the evaluation of the influence of plant extracts on the bacterial population adhering to the pellicle after treatment. The work is interesting in principle, since bacterial adherence has a central role in the formation of dental plaque. The methodology for the development of the work is appropriate and the experimental part has been well executed. The results are clearly showed, even though all speculations are limited due to the in vitro approach without in vivo tests. The length of the manuscript is in accordance with the results shown. The overall quality of the study is acceptable, but some minor criticisms arose during reading.
Even though I’m not the best person to judge the English style, the manuscript seems to be clearly written and comprehensive.

Minor points:
• Abstract. Several acronyms are reported. Some of them (MIC, FIC) are not specified. This is the first time they are cited and have to be reported in the extended form. It is very difficult to read an abstract without knowing what means what.
• Background. At the end of this paragraph, it should be reported the reason why it is important to investigate on the formation on biofilms of PEM.
• L100. MBC…specify.
• L132. “of for”.
• L204. Information on the drying process are necessary, since freeze drying is different from under vacuum concentration with regards of the residual antibacterial activities of plant extracts.
• L207-208. I doubt that a plant extract reconstituted (highly concentrated) could be easily filtered through 0.22 micron filters. Was the filtration performed before concentration?
• L217. How was it proven? By plate count? Please specify
• Tables 1, 2 and 3 report only a few data. They could be merged.

Experimental design

reported above

Validity of the findings

reported above

Additional comments

reported above

---

## Round 0.2 · accepted · Accept

· Academic Editor

Accept

Dear Authors,

Thanks for the submitted revised paper.

It has been greatly improved and it is now apt for publication.

Some minor suggestions are reported by the reviewer 1 which you can resolve in production

Best regards,
Maria Rosaria

Reviewer 1 ·

Basic reporting

No comments

Experimental design

No comments

Validity of the findings

No comments

Additional comments

The paper has been greatly improved and it is now apt for publication.
Some minor suggestions are reported in the following:
- l. 76 insert comma after the word "pellicle"
- l. 121 delete "in this study"
- l. 177 replace "a range of concentration" with "ranging", to avoid repetitions.
- l. 178 "was" is "were"
- Replace "ethical approval" after CONCLUSION and before ACKNOWLEDGEMENTS.

Reviewer 3 ·

Basic reporting

acceptable

Experimental design

acceptable

Validity of the findings

acceptable

Additional comments

acceptable